# Making Decision-Making Visible—Teaching the Process of Evaluating Interventions

**DOI:** 10.3390/ijerph18073635

**Published:** 2021-03-31

**Authors:** Angela Benfield, Robert B. Krueger

**Affiliations:** 1Occupational Therapy Program, Department of Health Professions, University of Wisconsin-La Crosse, La Crosse, WI 54601, USA; 2Entry-Level Doctor of Occupational Therapy, Whitworth University, Spokane, WA 97149, USA; rkrueger@whitworth.edu

**Keywords:** evidence-based practice, clinical reasoning, causal model, intervention theory, concept mapping

## Abstract

Significant efforts in the past decades to teach evidence-based practice (EBP) implementation has emphasized increasing knowledge of EBP and developing interventions to support adoption to practice. These efforts have resulted in only limited sustained improvements in the daily use of evidence-based interventions in clinical practice in most health professions. Many new interventions with limited evidence of effectiveness are readily adopted each year—indicating openness to change is not the problem. The selection of an intervention is the outcome of an elaborate and complex cognitive process, which is shaped by how they represent the problem in their mind and is mostly invisible processes to others. Therefore, the complex thinking process that support appropriate adoption of interventions should be taught more explicitly. Making the process visible to clinicians increases the acquisition of the skills required to judiciously select one intervention over others. The purpose of this paper is to provide a review of the selection process and the critical analysis that is required to appropriately decide to trial or not trial new intervention strategies with patients.

## 1. Introduction

All health professions have mandated the use of evidence-based practice (EBP) as a tenet of ethical practice [1] as it is known to improve healthcare quality, reliability, and client outcomes [2,3]. Over the last twenty years, skills and knowledge have significantly improved [4] and a great deal of effort (as well as the development of the fields of implementation science and knowledge translation) has gone into increasing the use of evidence-based practices (EBPs) by clinicians. However, these efforts have resulted in limited sustained behavior change for implementing EBPs [4,5]. A consensus of agreement suggests that appropriate implementation of EBPs is challenging due to a variety of factors. However, the theoretical “causes” of this is due to low EBP skill [4], organizational climate [6], and lack of “openness” to change practice [7]. However, there is no evidence that willingness to change practice is a causal factor as no studies have specifically measured the frequency of adopting new interventions; instead, most studies probe the adoption of specific interventions of interest (e.g., Locke et al. 2019 [8]). On the contrary, evidence suggests that openness to adoption of new interventions is not the problem as “fad” interventions continue to be reported in many areas of healthcare [9,10,11]. Most professionals report learning about new interventions from peers and rely on experience to select which ones to use in practice [12,13,14,15,16]. Importantly, it is the professional’s opinions on the intervention alternatives that most highly influence the client’s preference for action [17].

Second, we agree that some EBPs, such as handwashing for reducing infection, need to be implemented in all settings. However, not all interventions should be assumed to be the correct action in all settings with all clients. It is important to note that some EBPs can be adopted as written, some may need to be modified in a specific context or with a specific patient, and some will not be appropriate for adoption at all. Appropriate implementation of any given intervention is therefore dependent on many factors, including the confirmation that the clients are members of the same population as the sample from the intervention study being considered for adoption to practice. Frequently, the descriptions of whom the EBPs is directed provides insufficient information to determine if that the client is a member of the population [14,15]. Importantly, EBPs are often developed using evidence from randomized controlled trials (RCTs) [14]. However, the features of RCTs are such that increasing internal validity often leads to decreased external validity as appropriate application requires significantly more information about the participants, settings, and intervention than what is typically available in a published study. Registries (e.g., ClinicalTrials.gov) and manualized interventions are two strategies developed to address the generalizability difficulties of clinical trials. These strategies support the clinician in assessing generalizability by providing rich information on the participants, settings, and characteristics of the interventions used [18,19,20]. However, difficulties remain due to insufficient information [21]. Most interventions are complex, with many elements included in the intervention application. Many EBPs provide limited clear understanding of which intervention components are required activities (i.e., active ingredients) and which are not (i.e., inert ingredients). Therefore, partial implementation may result in the same outcome without doing all intervention components [22]. In other words, appropriate implementation of EBPs can be different across different settings.

Lastly, significant efforts and interventions have been applied in the last twenty years to increase EBP implementation rates. In fact, the main purpose of the field of implementation science is to identify the methods and strategies that facilitate evidence-based practice and research by clinicians [23,24]. Importantly, one of the main assumptions underlying implementation science research is that organizational climate affects implementation rates [6,25]. Therefore, strategies for increasing implementation of EBPs have targeted decreasing barriers, increasing supports, and increasing cultural expectations [6]. Although we agree these are important variables, we suggest an alternative causal mechanism should be considered. New evidence suggests that organizational climate is not a “causing” variable, but a “moderating” factor to EBP implementation [5]. Adoption of new interventions may not be the issue related to low EBP implementation as clinicians report frequently adopting and trialing new interventions with no or low evidence [16,26]. This suggests that we may need to take a step back and reexamine how we are defining and attempting to solve the problem of low implementation. We suggest that students need to be explicitly taught how to assess their personal causal models and the decision-making process.

## 2. Cognitive Mapping

Cognitive mapping has been used for over a half century to understand how learning changes a person’s cognitive structure of knowledge [27]. Initially, this instructional and assessment tool was used to understand meaningful learning in children. (Meaningful learning is defined as the learner has learned new information completely and has connected (identifying relationships) between new information and previously known knowledge [28]). Importantly, meaningful learning of the concepts and theory of a profession is the foundation of all professional knowledge [29]. Evidence supports the effectiveness of concept mapping to accurately capture a person’s knowledge and, further, can be used to measure change in knowledge [28]. The process of developing a cognitive map is consistent: first, the domain of knowledge is identified (e.g., a segment of text, a fieldtrip, a clinic, etc.); second, developing a specific question that the map will address sets the context; the third step is to use one’s tacit knowledge to identify key concepts. Some suggest to use a “parking lot” for key concepts where only factors that have an established relationship are placed in the cognitive map [27], while others suggest placing all the factors on the map, as that represents the individual’s understanding of the concept [30]. Once the concepts are on the map, the crosslinks between different factors are identified. It is the identification of these crosslinks of the underlying relationships between different factors that leads to the deeper understanding and the construction of new knowledge [27]. 

Cognitive mapping provides two avenues for improving evidence-based practices in healthcare. First, evidence on the effectiveness of cognitive mapping suggests that it is an effective tool to support organizing new knowledge into a coherent representation and integration with prior knowledge [31] and the development of clinical reasoning [32]. Second, cognitive mapping is effective in supporting divergent and creative thinking processes [33]. Divergent thinking is the ability to think originally, flexibly, and fluently [33]. Cognitive mapping encourages these activities by reducing the cognitive working memory load, which then frees cognitive capacity to engage in critical analysis and problem solving, e.g., identify associations, search for alternative perspectives, etc. [29,31,33]. Evidence also supports that it helps learners engage in dynamic thinking (not linear thinking) [34], the holistic analysis of the concepts or problems, and leads to new cognitive representations of the problem (or concept) [33]. In other words, the outcome of the cognitive mapping process supports adaptive change through changing their cognitive representations of the problem, which supports identifying new solutions to “old” problems. Importantly for an EBP implementation, cognitive mapping, specifically fuzzy cognitive mapping, has been identified as an effective tool to understand complex, uncertain problems and the perceptions (e.g., causal functions of factors) of the various stakeholders [35]. Evidence supports that cognitive mapping of complex problems is able to identify the causal structures used by various stakeholders to frame the problem and influencing the action decision [35], and can affect achieving organizational change [36,37].

## 3. EBP Curricula

We agree that the development of various EBP curricula has led to better skills and knowledge of how to appraise and synthesize evidence to support local practice. These curricula have taken a complex, multistep process and made it easier by directly teaching the specific action steps that the clinicians are expected to do (e.g., Ask, Identify, Appraise, Apply, Evaluate). Further, appraisal systems and tools (e.g., GRADE, CASP, Cochrane) decrease cognitive load by providing cues or reminders on questions that need to be answered. More elaborate systems, like GRADE, even provide decision trees to decrease the chance of making an error. Importantly, the EBP process, appraisal systems, and decision-support tools have been demonstrated to improve the positive impact on the quality of the care and clinical outcomes as they increase the likelihood of selecting and enacting the correct decision [1]. However, the decision-support tools typically are only available to address clinical questions where there is more certainty (e.g., differential diagnosis decision-making of known problems, depression medication choice, statin choice (Available online: https://carethatfits.org/ (accessed on 14 February 2021)), and sufficient evidence to develop EBP activities. Critically, many of the intervention decisions are made in uncertain conditions [1], which lack strong, unequivocal evidence that there is only one correct action.

## 4. Problem Solving in Medicine

Since the Renaissance, science and medicine were rooted in the assumption that if problems are divided into smaller and simpler units, they become easier to solve [38]. These smaller units lent themselves to linear causality, e.g., one cause produces one effect [39] and heavy reliance on linear diagrams and flow charts [40]. Recently, support has been voiced for using “holistic” or biological system approaches to think about clinical problems; however, many technological advances continue to require increased specialization that reinforce reductionistic approaches [41]. On the other hand, understanding a problem from a dynamic systems approach requires one to examine it from a multidisciplinary approach leading to the mandate of interprofessional care [42]. However, this same push introduces another level of complexity as each specific discipline advocates for their individualized profession’s scope of practice, allocation of resources, and different professional definitions of the problem, goal, action steps, and outcome indicators [43]. Modern healthcare services are consistent with the blind men and elephant parable. Each profession uses their own professional paradigm, experiences, and client interactions to frame and solve the client problem. Importantly, it is more likely that each professional will have a different causal model that they are using to organize their knowledge and examine the problem [44]. In fact, even members of the same profession are just as likely to have different mental conceptualizations of the same client that may not be supported by the evidence or formal theories (e.g., professional paradigms) [45,46]. Critically, integrated knowledge has been implicated as prerequisite for successful problem solving [47,48] and evidence supports it is the restructuring of knowledge as new information is learned leads to richer causal models, which leads to being more likely to select appropriate actions [49].

Evidence on what humans do when presented with a novel problem suggests that heuristic reasoning is used in time limited situations [50], as most people require approximately eleven to sixteen seconds to process, interpret, and formulate an initial reflective response [51]. In medicine and healthcare, heuristics is a strategy that is used when decisions need to be made quickly, as it ignores information as a tradeoff between time available and the cost of getting better information [52]. Fast thinking can be highly influenced by cognitive biases and include ambiguities [53,54] Actions taken in the past are easier to remember than the actions not taken, [55] or how the problem was initially defined. Importantly, fast thinking (e.g., intuition, heuristics) also highly influence initial seeking and interpretation of information, but also early closure if purposeful conscious thinking efforts are not engaged [56,57]. One common educational intervention frequently used to slow down closure has been the use of “think aloud” during which the respondent describes their evidence, making their thinking visible to communication partners [50]. This allows all involved to have more time to reflect on the quality of the judgment and to compare evidence to their own reflective thinking process [50]. However, “think aloud” was not a solution for all difficulties. Reflective practice and EBP frameworks were developed to best address limitations in decision-making and heuristic reasoning. Both strive to increase critical reflection and integration of nonexperiential evidence into the decision-making process [56]. There is general consensus that thinking changes with experience and expertise [58,59,60,61,62,63,64,65,66,67]. One benefit of this change is that experienced clinicians often recognize similar situations (e.g., familiar problems) more quickly [68]. Importantly, experience may not improve their ability to adapt commonly used strategies to resolve new problems or increase their ability to recognize errors in their own thinking processes [50,69,70,71]. Since heuristics will always have a role in decision-making due to the speed expected in real-time interactions, it is important to understand what information is used in heuristic decision-making.

### 4.1. Heuristics and Causal Models

In everyday life, adaptive response is the outcome of rapid decisions based on numerous different interrelated variables [72,73]. In any given situation, we identify the variables which we believe are “causing” or influencing the current situation [72,73,74]. In other words, decision-making is not linear, but a dynamic process, where there are many alternative options for achieving an outcome [72]. Interestingly, there is strong consensus that the clinician’s beliefs about clinical problems are probabilistic and are weighted in the causal model by how likely any individual variable is believed to be causing or influencing the outcome [1], consistent with Bayesian inference [75]. Importantly, these mental models represent a person’s theory of how the world works and also include estimations of their probabilistic relationship of the variable to the situation and the desired outcome [38,73]. Clinicians use their mental model to select the actions that their personal theory posits as the most likely to achieve the desired outcome given a specific situation. Significant evidence supports that it is this mental model which drives intuitive and heuristics decisions [76,77,78,79]. Deliberate practice and EBP were specifically targeted to support clinicians in solving problems as they emerge. We agree that these models for supporting practice are important and should not be discounted. However, professional education curricula need to directly teach students how to think about their thinking, appraise their knowledge, and how to integrate of information into rich evidence-informed causal models. Importantly, we also posit that by clearly teaching this skill as an expectation for ethical practice, completed (at least informally) when new information is learned, it significantly increases the expectations of how frequently they should engage in these activities.

### 4.2. Causal Models and Interventions

Ideally, each individual clinician relies on their own causal model to select the interventions to use with a specific client [80]. Causal models develop over time as they are the individualized outcome of the clinician’s thinking process, synthesizing formal professional theories, lived experiences, and overlying the client’s preferences and values related to the problem. Causal models use “rules” to identify the relationship between the factors, problem, and outcomes that are achievable. The rules develop from professional paradigm, experience, and tacit knowledge. The accuracy of the rules (and their causal model) will depend on how the clinician represents the problem, the number of factors identified, and the factors that are missing. Evidence supports that people’s causal judgments can be influenced by observed data when the data are consistent with their mental model; this thinking process is highly influenced by limited knowledge, missing information or ignored causal assumptions, and cognitive biases [4]. However, making this causal model visible and providing specific strategies (e.g., seeking evidence that the factor is influencing the problem and remediable) increases the likelihood that the causal model correctly weighs the factors that are really causing the problem [11,17,73].

### 4.3. Causal Mapping

Linear thinking leads to erroneous causal assumptions, as it relies on the readily available data points (e.g., proximal goal attainment), has difficulty accounting for feedback, and has difficulty accounting for other variables that do not fit in the linear model (e.g., “what if X causes Y, rather than Y causes X, or Z causes Y and X”) [6,42]. Historically, cognitive maps have been used with subjects who have extensive knowledge; however, cognitive maps also have the ability to provide insight into how new information is integrated and synthesized with previous experience and knowledge [3]. Cognitive maps have been used extensively in education as a teaching tool, as they reduce cognitive load [50] and encourage deeper learning of concepts by supporting the identification of the relationships between variables [80]. Empirical evidence finds that cognitive maps help students organize the information into a format that is retained at higher levels, and allows the learner to encode the information in both visual and language forms (e.g., conjoint retention) [80]. The visual representation of complex relationships is easier to follow compared to relationships between variables described by words alone, and they foster critical thinking and reasoning [80,81]. McHugh Shuster (2016) provides an example of how concept mapping helped nurses organize patient care by organizing the patient data, analyzing relationships, establishing priorities, builds on previous knowledge, and encourages a holistic view of the client [82]. Importantly, cognitive mapping has been demonstrated to support EBP uptake [83,84]. This may be due to integration of the new information into their causal model of the problem [84], which increases the likelihood of an EBP being viewed as an alternative action to achieve the outcome. Importantly, evidence also provides an understanding of how linguistics and analogies, including testimonials, moderate and change an individual’s causal model [39], highlighting a critical method that could be used to shift causal models.

### 4.4. Making Decision-Making Visible

The decision of selecting one intervention over others arises out of the intersection of two different critical analysis processes: (1) the personal causal model of the problem and (2) the analysis of interventions (refer to Figure 1. Simple view of intervention selection). The outcome of this process is to select, from all plausible interventions, the one that the clinician believes is the most likely to achieve the outcome given the specific situation and to be able to identify the level of evidence (empirical or theoretical) and confidence of the decision.

Many of the interventions commonly used in practice have limited evidence of effectiveness [85] and are complex with multiple moderating and affecting variables [86]. The goal of this process is to develop an evidence-informed causal model of the clinical problem, which allows the student (or clinician) a deeper and more complex understanding of the theorized causal relationships of the variables. Therefore, this process brings awareness to what, how, and why individuals are thinking. Importantly, this process expands the reasons that one seeks new information as one main purpose is to advance the currency of knowledge with clinical problems, which they may address in the future, and expand the integration of knowledge from various sources. The use of expected routines and habits for thinking about practice is from deliberate reflective practice [87,88] where clinicians respond to triggering events through a structured process of reflection and critical analysis. In contrast, evidence-informed thinking is an expected habit of practice done frequently and routinely—not just when triggering events occur.

In making causal models visible, students are taught the first step when confronted with a problem (e.g., medical condition, clinical problem, culture, spirituality) is to tentatively define the concept or problem and then critically reflect and analyze what they think they know, believe, and value. At this stage, they use their intuition and tacit knowledge to identify variables that they think may be involved. Consistent with the brainstorming stage of problem-solving frameworks, variables are placed on the map without regard to strength or plausibility of its relationship, as the goal is to bring to consciousness the variables that may influence thinking at later stages. In the second stage, they are explicitly taught to seek and read the literature and talk with peers in order to identify other variables that are theorized or empirically known to be influencing the problem. They add the newly identified variables and add linking verbs to identity relationships between the variables (theoretical or supported by evidence). At this stage, they also indicate the strength and direction of the relationship (one-way, two-way, or related through other variables) [89]. At this stage, some authors suggest removing concepts that have no relationship [89]. However, there may be benefits to leaving all variables on the map and including the evidence that shows the lack of relationship. This may provide a conscious, visible “nudge” to reflect on why they believe there is a causal relationship and deeper analysis of the phenomena. Importantly, the “lack of evidence” may be due to the lack of data on the relationship, not the lack of a relationship; this, therefore, supports the need for identification and collection of local data to use in outcomes reflection (refer to Table 1. Making thinking visible—evidence-informed causal model). Importantly, this process is needed to support the deliberate practice models; however, deliberate practice models are not fully described here for simplicity purposes. 

Making thinking visible—evidence-informed causal model.

For clinical problems, causal maps provide four specific benefits. First, it makes searching for evidence feasible and more efficient as the variables can become search terms. Second, the factors can be used to identify interventions that are theorized to affect the clinical presentation and the outcome of interest. Third, a causal model in which interventions have been added will contain alternative options for achieving an outcome. Interventions connected to different variables can be inserted into the clinical question (e.g., Population, Intervention, Comparison, Outcome (PICO). In other words, the variables and concepts included on a cognitive map provide starting points for the formal search strategies found in EBP textbooks, which can be a very difficult and frustrating experience for people with limited knowledge of a phenomena. Figure 2, Figure 3 and Figure 4 provide an example of this process as it relates to handwriting dysfunction in young children. Finally, this process supports the development of evidence-informed causal models and the integration of new information into tacit knowledge by having them add new information as they go through their professional programs and link different cognitive maps together. Using cognitive mapping is consistent with spiral curricula. In spiral curricula, there is a purposeful revisiting of topics whereby each revisit builds on previous knowledge by both deepening their knowledge (and complexity of understanding), but also expanding the connectedness to other knowledge [90]. Using tools such as CMapTools^®®®^ (Available online: https://cmap.ihmc.us/ (accessed on 14 February 2021)) allows for students to actively organize and construct their own maps, which also allow them to connect maps together, embed resources, and share maps with others. 

As illustrated in the figures, cognitive maps are able to cogently translate information and complex relationships. 

### 4.5. Intervention Theory

Many interventions addressing behavioral change are complex, consisting of various components that have been combined as they theoretically cause change to important variables affecting the clinical presentation [91]. Intervention theory consists of three components: the target that the intervention changes, the mechanism of change, and the components.

Components, or “ingredients” in intervention theory, include anything that is done by the clinician to a client, including, for example, medications, assistive devices, motivational speech, ultrasound, manual touch, instruction, feedback, and physical cues. Ingredients target small aspects of the clinical outcome. In practice, interventions contain both active ingredients that are required to enact change in outcome, and inert ingredients, which have no effect on the targeted outcome [92,93]. Identifying the active ingredients can be cost-effective as it reduces time and resource requirements because fewer activities need to be done in order to achieve the outcome [22]. Typically, there is not a clear understanding of which ingredients of an intervention are active [93,94]. Another reason is that there is a lack of clear identification of all components being applied in the intervention. This is likely due to the difficulty in separating the complex relationships between the components when change occurs in dynamic systems [91]. In other words, the form of delivery (when, where, how, how much, who, etc.) is critical to the “active ingredient” [95]. Therefore, being able to assess intervention fidelity is critical. Fidelity is “the extent to which the core components of a program, differentiated from “business as usual,“ are carried out as intended upon program enactment” [22] (p. 320). Fidelity in studies provides the confidence that it was the intervention that caused the desired effect. Fidelity measures can also be used to differentiate those components more likely to be active ingredients from those that are inert [22]. This is done by comparing the percentage of the components being delivered by the components as defined, and those that are not; this allows assessment of the impact the outcome effects by each component. If the fidelity is low, but equal affects are achieved, the active ingredient is more likely to be one that was causing the change [22]. 

An example from the clinic on how fidelity tools were used to examine practice is the history of constraint-induced movement therapy (CIMT). CIMT is an intervention that addresses function of a hemiplegic upper extremity. Early protocols called for constraining the uninvolved upper extremity for 90% of the waking hours and intensive therapy for six hours a day, for 10 days over two weeks [96]. However, current understanding of the effects of various treatments for increasing affected extremity function use suggest that CIMT is no more effective than high-dose standard care occupational therapy or dose-matched bimanual therapy, neither of which the “constraining” ingredient is included [97]. This is a great example where assessing local application fidelity to the original protocol allowed clinicians to identify differences in the application of ingredients (including dose, timing, etc.). When consistent outcomes were achieved, it led to significantly reduced time of intensive therapy, using oven mitts (versus expensive “therapy” mitts), and eventually to examining alternative protocols such as bimanual training [98]. Currently, the evidence suggests that these interventions (CIMT, bimanual, and intensive standard care) work due to the dose of the active ingredient, task-specific training [99]. Importantly, research on task-specific training is currently exploring more specific questions (e.g., dose, timing) for which critical information is needed to develop effective (and cost-effective) nonburdensome, interventions [100]. 

One issue with many EBPs is that the descriptions of the interventions lack a clear explanation of parameters (i.e., what, when, how, or how long) for any specific activity or component [95,101]. This “black box” of intervention components creates a barrier for translation to different contexts and clients [14,101,102,103] and explains the commonly reported dependence using peers and what was previously done for selecting interventions [14,101]. Treatment theory has been proposed as a tool to make intervention selection clearer, and make outcome data more interpretable and more transferable to different clients and contexts [92,94,103]. This process consists of three structures—the *target of the treatment* (e.g., body function, structure, participation) that is altered by the intervention; *the ingredients* that produce the change; and *the mechanism* that the ingredients cause change in the target of the treatment [92,94,102]. This analysis, as well as traditional appraisal of estimates of effect, is what allows for appropriate application of an intervention with a different client and setting [92,103] (see Figure 5. Three-part structure of intervention and intervention selection).

### 4.6. Teaching How to Think about Interventions

We argue that it is important to explicitly teach students what information they should seek when learning about a new intervention. Due to the complexity of information that impacts appropriateness, feasibility, and applicability, the quality indicators for interventions have been identified using suggestions of best practice from a variety of areas: intervention theory, total quality improvement, continuing education effectiveness, clinical reasoning, and EBP. Interestingly, there were similarities across the literature. Appropriate application and increased likelihood of actually selecting an intervention depends on first and foremost, the learner gaining an understanding of the problem (e.g., defining) and the need for better client outcomes—in other words, what it typically achieved is less than the potential [91,94]. This provides the motivation that they need to seek and learn about new interventions, engaging the potential to change behavior [91]. Second, appropriate application is dependent on the learner gaining information about the problem: what interventions (and active components) work, why they work, and any contextual factors that may influence the effectiveness of the intervention [1,2]. Contextual factors are critical, including what components were done, when, where, how, and how much [94]. It also includes a discussion of alternative interventions that would be used to weigh plausible different actions [92,102] and the measurement indicators that were used to support differentiation [94,103]. Importantly, establishing the habit of discussing commonly used alternatives can help the learner access their experiential knowledge, which can be used to compare client factors, activities, etc. [2]. By having the students use the intervention rubric to reflect on the information provided (from any source) on a specific intervention reduces the cognitive load and encourages asking (seeking) for needed information. It also supports the student’s process of identifying when there is a lack of evidence (empirical or theoretical), lack of causal mechanism, and/or lack of clearly identifying data points that would indicate change (see Appendix A. Intervention rubric).

#### Clinical Example of Intervention Analysis

By the mid-1990s, auditory integration training became a worldwide popular intervention for people with autism. It was theorized to improve many personal factors such as attention span, eye contact, tantrums, etc. Due to its prevalent use, multiple efficacy studies were completed and showed that the intervention had no effect [104]. However, by the time the Cochrane Collaboration issued its findings, the original intervention, attributed to Berard (1993) [105], had been modified and adapted into multiple interventions with new names (e.g., Tomatis method (Available online: https://www.tomatis.com/en (accessed on 12 February 2021)), (Available online: https://soundsory.com/samonas-sound-therapy/ (accessed on 12 February 2021)), Listening Program (Available online: https://advancedbrain.com/about-tlp/) (accessed on 12 February 2021)), Therapeutic Listening (Available online: https://vitallinks.com/therapeutic-listening/ (accessed on 12 February 2021)), etc.) [106]. Importantly, it is only through an analysis of the critical aspects of the interventions that a clinician becomes aware that the new interventions use the same treatment theory. Differences (e.g., name, type of headphone, location to purchase music, etc.) appear to be only surface changes. Therefore, the evidence as appraised by Sihna et al. (2011) would suggest that all of these new interventions would also lack of evidence effectiveness. Importantly, these new iterations of auditory integration intervention continue to be used by clinicians, although the developer now targets a different audience (e.g., occupational therapists, educators, and parents). Interestingly, the information readily available on the interventions is limited but located on highly attractive websites that present powerful narratives of client change by the people who have financial incentives to sell the tools of the intervention. Importantly, none of the newer iterations clearly provide specific examples that support viewing them as having a different theory of causal mechanism from the studies that showed no effect [106].

## 5. Conclusions

In order to select the best intervention, it is essential that the clinician use an evidence-informed causal model and has appropriately integrated information on many different variables. Evidence supports that making the complex thinking process visible supports deep learning and improves the causal reasoning, which is used to predict the best intervention option. Expanding our instructional methods to include direct instruction (making thinking visible) has the potential to improve the accuracy of decisions, and therefore, client outcomes, as it supports the development of underlying thinking habits of clinicians. Importantly, intervention selection and EBP are both highly influenced by the clinician’s causal model. Therefore, improving the quality and habits of thinking has the potential to increase the likelihood of selecting the “best” interventions with our clients. Specifically, teaching and helping students to establish the habit of developing an evidence-informed causal model has the potential to improve heuristic decision-making as evidence suggests that one’s causal model highly influences heuristic decision-making. Using cognitive support tools such as the intervention rubric (Appendix A) also has the potential to improve appropriate selection of interventions by encouraging deeper analysis of interventions and identifying measurement data points that can be used to assess the accuracy of the information used to select it. Using cognitive mapping, specifically causal modeling, also supports the EBP process by highlighting alternative terms and interventions strategies to use in the EBP process. 

## Figures and Tables

**Figure 1 ijerph-18-03635-f001:**
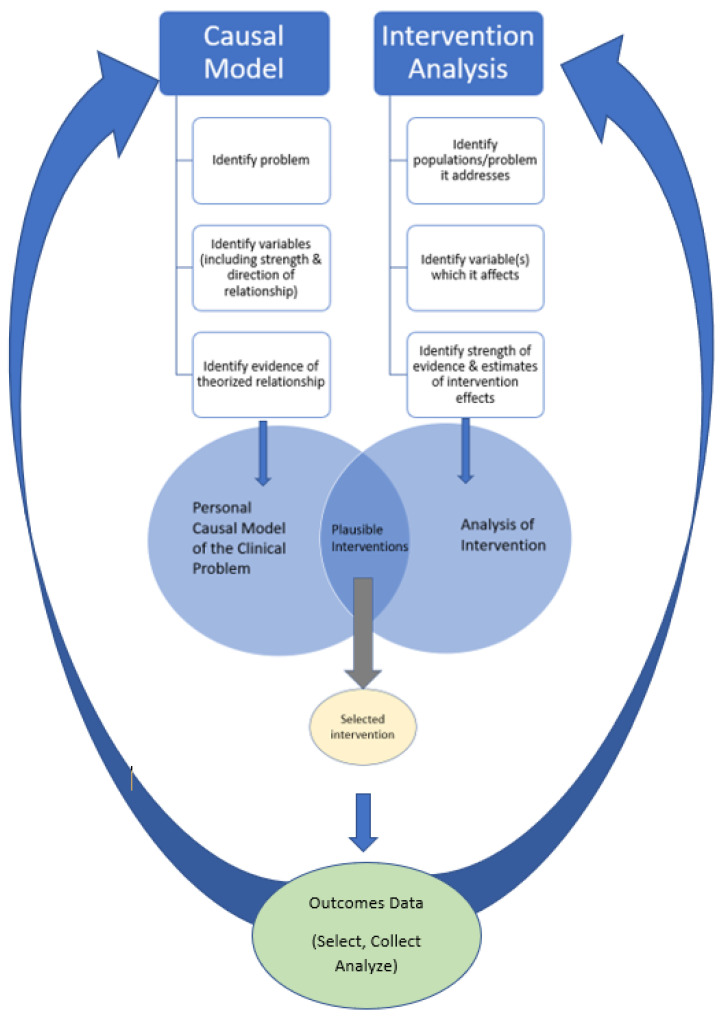
Simple view of intervention selection (diagram only contains the main steps of each process).

**Figure 2 ijerph-18-03635-f002:**
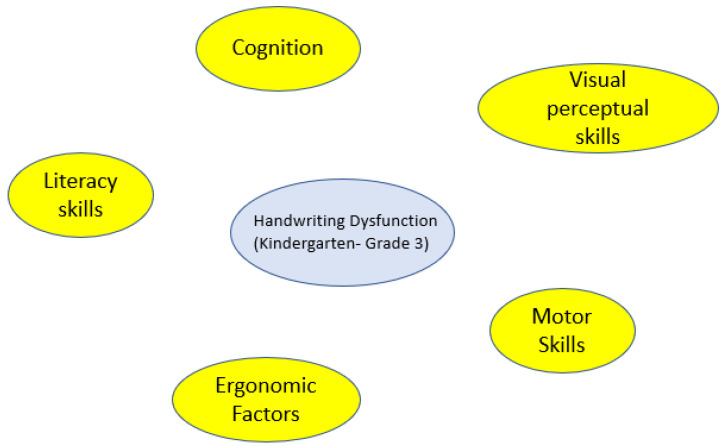
Intuitive factors thought to affect handwriting dysfunction.

**Figure 3 ijerph-18-03635-f003:**
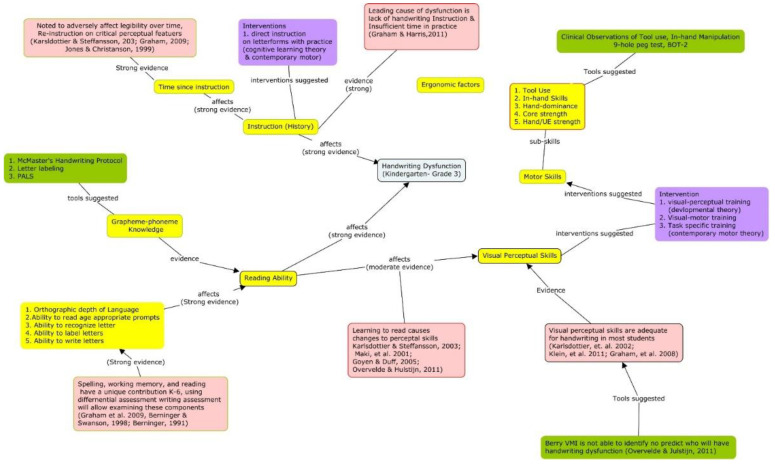
Preliminary causal model after searching the literature. It includes direction of causal relationship and strength of evidence supporting the relationship. Placing the evidence provides ability to confirm appraisal of evidence.

**Figure 4 ijerph-18-03635-f004:**
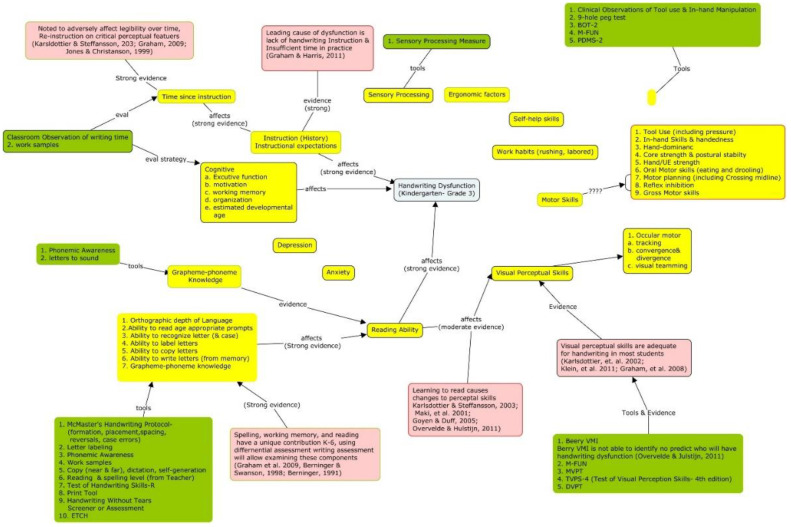
Evidence-informed causal model with interventions and assessment tools suggested in the literature. Blue indicates the target problem. Yellow indicates factors which are believed, theoretically and/or empirically to have a relationship to the target outcome. Pink indicates evidence on the factor’s relationship to other factors or target problem. Green indicates assessment tools.

**Figure 5 ijerph-18-03635-f005:**
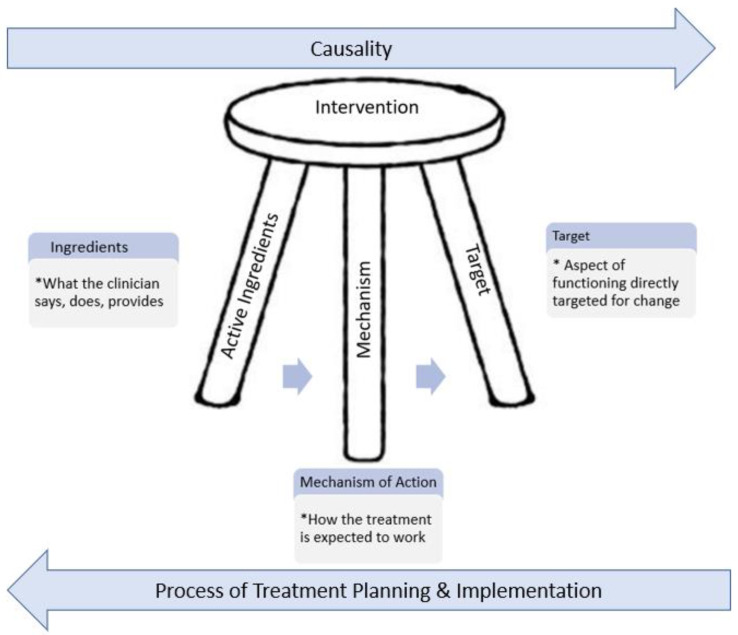
The Three-part structure of Intervention and Intervention Selection. Based on [83].

**Table 1 ijerph-18-03635-t001:** Making thinking visible- Evidence-informed causal models.

	Action
Step 1	Identify possible problem(s), concepts, etc.
Step 2	Reflect on your beliefs, knowledge, experiences, and place any variable which may be affecting the problem on the map
Step 3	Seek empirical evidence and background information (a) relationship between the factor, the problem, and the outcome (b) a plausible explanation of how that factor “causes” the problem (c) evidence that the factor is remediable
Step 4	Identify and link interventions to the variables they affect
Step 5	Assess applicability, feasibility to context and client
Step 6	Identify and collect outcomes data points locally which will allow assessing accuracy of decision
Step 7	Reflect on outcomes data: (a) assess impact of intervention (b) assess accuracy of the action taken (assess clinical judgment) (c) assess accuracy of causal model

## Data Availability

There is no data.

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
