# Peer review of "Making Decision-Making Visible—Teaching the Process of Evaluating Interventions"

_ijerph, 2021, doi:10.3390/ijerph18073635_

Round 1
Reviewer 1 Report
This is an interesting paper which presents a comprehensive discussion on challenges with teaching concepts on evidence implementation. I agree this is an area that is often not considered in-depth when teaching EBP and there are often several reasons for this.
The paper is currently presented as a literature review rather than a systematic or scoping review (or other formal review process e.g. using PRISMA guidelines or Cochrane, JBI methods). As this paper is therefore not an empirical research study, nor is it reported using a rigorous framework as mentioned above, it is difficult to identify the evidence behind the proposed model.
There are some generalizations made in the introduction which require stronger justification or change in wording e.g. line 31 “All agree” could be changed to “Research suggests…” or “ Consensus of opinion suggests…” In fact the challenge with papers such as these is to ensure a balanced view is presented throughout.
It would be good to see the paper reframed and narrowed in scope, using a recognized methodology to provide a more rigorous report. This would also provide clarity on the aims and objectives of this paper as well as the utility of the conceptual model.
Additionally, there have been other papers published which may be worth considering for supporting evidence e.g. Phillips, A.C., Lewis, L.K., McEvoy, M.P. et al. Development and validation of the guideline for reporting evidence-based practice educational interventions and teaching (GREET). BMC Med Educ 16, 237 (2016). https://doi.org/10.1186/s12909-016-0759-1
Albarqouni, L., Hoffmann, T., & Glasziou, P. (2018). Evidence-based practice educational intervention studies: a systematic review of what is taught and how it is measured. BMC medical education, 18(1), 1-8.Aglen, B. (2016).
Young, T., Rohwer, A., Volmink, J., & Clarke, M. (2014). What Are the Effects of Teaching Evidence-Based Health Care (EBHC)? Overview of Systematic Reviews. PLOS ONE, 9(1), e86706.
Pedagogical strategies to teach bachelor students evidence-based practice: A systematic review. Nurse education today, 36, 255-263.
Theoretical based interventions for behaviour change have also been reported on in several systematic reviews which may be worth consideration:
Davis, Rachel, et al. "Theories of behaviour and behaviour change across the social and behavioural sciences: a scoping review." Health psychology review 9.3 (2015): 323-344.
Godin, G., Bélanger-Gravel, A., Eccles, M., & Grimshaw, J. (2008). Healthcare professionals' intentions and behaviours: A systematic review of studies based on social cognitive theories. Implementation Science, 3(36), 1-12.
Kwasnicka, D., Dombrowski, S. U., White, M., & Sniehotta, F. (2016). Theoretical explanations for maintenance of behaviour change: a systematic review of behaviour theories. Health psychology review, 10(3), 277-296.
Johnson, Mark J., and Carl R. May. "Promoting professional behaviour change in healthcare: what interventions work, and why? A theory-led overview of systematic reviews." BMJ open 5.9 (2015): e008592.
Author Response
We have been revised our manuscript according to your specific comments. Please check if the revision meets your requirement.
We thank the reviewer for the hard work and patience!
Reviewer 2 Report
In my opinion, this is a work of huge interest in the field of implementation sciences and in the framework of knowledge translation, because it attempts to analyze the apparent “failure” (or very limited success) of the different methods and strategies used to date to achieve systematic implementation of the results of scientific research (EBPs) in different types of organizations, and also puts forward other points of view on the subject and alternatives to approach it, preferably from the educational environment.
I think it is only necessary to review the work so as to correct various spelling and writing aspects.
Some spelling errors detected:
- Line 53 page 2: "cllinicalgov"
- Lines 218-220 page 6: the title of Table 1 seems to be repeated in these two sentences. Please review.
- Table 1 (Step 7, section b). A punctuation mark seems to be missing (please correct the sentence).
- Figure 3. There are spelling mistakes: "uniquecontribution", "diffenential". Please review
- Figure 4. Correct the spelling errors already detected in Figure 3. Define the acronyms that appear (e.g., PALS). To improve understanding, I think it would be interesting to identify at the bottom of the figure what each color code used corresponds to (e.g., purple-intervention / s, ... etc.
- Line 249 page 7: "ingredients".
- Line 258 page 7: “casused”.
- Line 275 page 275: is the phrase “… research on task specific training research…” correct? Please, correct if required.
- Line 279 page 8: A punctuation mark is missing before "Of"
- Line 311 page 8: The section number must be wrong "3.5.1".
- Line 210 page 6 refers to a “third stage”. However, in the text there is no explicit mention of a “second stage”. Please review and correct, if required.
- In the "Reference" section:
- There are some references with apparently incomplete information (e.g. ref 28, 39, 40, 57, 67…). Please review all the other bibliographic references to check their completeness.
- Please also check the missing punctuation marks (e.g. between the titles of the articles and the journals that publish them).
- Some journal names appear with the full name, instead of with the abbreviated name (e.g., ref 2, 47, 48, 54, 55…).
In short, please review the full “References” section to adapt to the standards established by the journal and with a uniform structure for each of the types of reference (journal articles, book and book chapters…).
Author Response
Point 1:
I think it is only necessary to review the work so as to correct various spelling and writing aspects.
Some spelling errors detected:
- Line 53 page 2: "cllinicalgov"
- Lines 218-220 page 6: the title of Table 1 seems to be repeated in these two sentences. Please review.
- Table 1 (Step 7, section b). A punctuation mark seems to be missing (please correct the sentence).
- Figure 3. There are spelling mistakes: "uniquecontribution", "diffenential". Please review
- Figure 4. Correct the spelling errors already detected in Figure 3. Define the acronyms that appear (e.g., PALS). To improve understanding, I think it would be interesting to identify at the bottom of the figure what each color code used corresponds to (e.g., purple-intervention / s, ... etc.
- Line 249 page 7: "ingredients".
- Line 258 page 7: “casused”.
- Line 275 page 275: is the phrase “… research on task specific training research…” correct? Please, correct if required.
- Line 279 page 8: A punctuation mark is missing before "Of"
- Line 311 page 8: The section number must be wrong "3.5.1".
- Line 210 page 6 refers to a “third stage”. However, in the text there is no explicit mention of a “second stage”. Please review and correct, if required.
- In the "Reference" section:
- There are some references with apparently incomplete information (e.g. ref 28, 39, 40, 57, 67…). Please review all the other bibliographic references to check their completeness.
- Please also check the missing punctuation marks (e.g. between the titles of the articles and the journals that publish them).
- Some journal names appear with the full name, instead of with the abbreviated name (e.g., ref 2, 47, 48, 54, 55…).
In short, please review the full “References” section to adapt to the standards established by the journal and with a uniform structure for each of the types of reference (journal articles, book and book chapters…).
Response 1: All spelling, punctuation, and reference errors have been corrected. References were reformatted.
Reviewer 3 Report
See attached file.

Author Response
Point 1: Minor Points
1. l. 65 of science ... the organizational
2. l. 97 On the other ...
3. l. 211 to be ináuencing
4. l. 212 which speciÖcally identify
5. l. 249 ingredients (misspelled twice) also on l. 252
6. l. 325 by the very people
Response 1: All have been corrected.
Reviewer 4 Report
General Remarks:
- The overall text is carefully written and very clear, as far as the theoretical framework is concerned. I understand the paper’s aim is fundamentally expository; however, I believe that if the idea is to improve teaching methodologies concerning complex thinking processes that may support EBP, empirical data would reinforce authors beliefs.
- I also think that the concept of concept map should be more developed.
- Examples of concept maps (CMs) designed by students materializing complex thinking processes and causal reasoning concerning interventions could illustrate the cognitive potential of CMs as tools for organizing and representing knowledge (Novak & Cañas, 2006).
- As for the term client, the word seems fuzzy to me. Why is it more suitable than patient?
Specific Remarks:
Line 53 – *Cllinicaltrials.gov) – There is a typing mistake, I believe.
Line 241 - As for format, the font is not the same after line 241 (3.5).
Lines 248 to 252 - * ingrediants ? *Ingredigants? The words seem to be misspelled
Author Response
Point 1: The overall text is carefully written and very clear, as far as the theoretical framework is concerned. I understand the paper’s aim is fundamentally expository; however, I believe that if the idea is to improve teaching methodologies concerning complex thinking processes that may support EBP, empirical data would reinforce authors beliefs.
Response 1: Evidence on the effectiveness has been added to the paper.
Point 2: I also think that the concept of concept map should be more developed.
- Examples of concept maps (CMs) designed by students materializing complex thinking processes and causal reasoning concerning interventions could illustrate the cognitive potential of CMs as tools for organizing and representing knowledge (Novak & Cañas, 2006)
Response 2: background information on cognitive maps has been added. However, a full discussion was not provided as the purpose of the paper is to provide instructors of professional programs that this effective tool is applicable to improve deeper learning and the construction of new/modified cognitive models of knowledge which underlie providing ethical care
Point 3: As for the term client, the word seems fuzzy to me. Why is it more suitable than patient?
Response 3: The term client is commonly used across the world as many healthcare practitioners no longer work in traditional medical settings. Client designates that the receiver of the health services has a choice to obtain services from other providers- It also designates that the client is a person who seeks assistance and is an active partner in the decision-making to improve their life.
Point 4: grammar, mechanics, etc.
Response 4: All grammar, mechanics, font changes, reference formatting has been addressed.